# Lifestyle and Pain following Cancer: State-of-the-Art and Future Directions

**DOI:** 10.3390/jcm11010195

**Published:** 2021-12-30

**Authors:** Astrid Lahousse, Eva Roose, Laurence Leysen, Sevilay Tümkaya Yilmaz, Kenza Mostaqim, Felipe Reis, Emma Rheel, David Beckwée, Jo Nijs

**Affiliations:** 1Research Foundation—Flanders (FWO), 1000 Brussels, Belgium; Laurence.leysen@vub.be; 2Department of Physiotherapy, Human Physiology and Anatomy, Faculty of Physical Education and Physiotherapy (KIMA), Vrije Universiteit Brussel, 1090 Brussels, Belgium; Eva.Charlotte.S.Roose@vub.be (E.R.); sevilay.tumkaya.yilmaz@vub.be (S.T.Y.); kenza.mostaqim@vub.be (K.M.); Emma.Rheel@vub.be (E.R.); David.Beckwee@vub.be (D.B.); jo.nijs@vub.be (J.N.); 3Pain in Motion Research Group (PAIN), Department of Physiotherapy, Human Physiology and Anatomy, Faculty of Physical Education and Physiotherapy (KIMA), Vrije Universiteit Brussel, 1090 Brussels, Belgium; felipe.reis@ifrj.edu.br; 4Rehabilitation Research (RERE) Research Group, Department of Physiotherapy, Human Physiology and Anatomy, Faculty of Physical Education and Physiotherapy (KIMA), Vrije Universiteit Brussel, 1090 Brussels, Belgium; 5Physical Therapy Department, Instituto Federal do Rio de Janeiro (IFRJ), Rio de Janeiro 20270-021, Brazil; 6Postgraduation Program, Clinical Medicine Department of Universidade Federal do Rio de Janeiro (UFRJ), Rio de Janeiro 21941-901, Brazil; 7Department of Experimental-Clinical and Health Psychology, Ghent University, 9000 Gent, Belgium; 8Department of Physical Medicine and Physiotherapy, University Hospital Brussels, 1090 Brussels, Belgium; 9Unit of Physiotherapy, Department of Health and Rehabilitation, Institute of Neuroscience and Physiology, University of Gothenburg, 405 30 Gothenburg, Sweden

**Keywords:** cancer survivor, chronic pain, lifestyle, diet, obesity, physical activity, stress, sleep

## Abstract

This review discusses chronic pain, multiple modifiable lifestyle factors, such as stress, insomnia, diet, obesity, smoking, alcohol consumption and physical activity, and the relationship between these lifestyle factors and pain after cancer. Chronic pain is known to be a common consequence of cancer treatments, which considerably impacts cancer survivors’ quality of life when it remains untreated. Improvements in lifestyle behaviour are known to reduce mortality, comorbid conditions (i.e., cardiovascular diseases, other cancer, and recurrence) and cancer-related side-effects (i.e., fatigue and psychological issues). An inadequate stress response plays an important role in dysregulating the body’s autonomic, endocrine, and immune responses, creating a problematic back loop with pain. Next, given the high vulnerability of cancer survivors to insomnia, addressing and treating those sleep problems should be another target in pain management due to its capacity to increase hyperalgesia. Furthermore, adherence to a healthy diet holds great anti-inflammatory potential for relieving pain after cancer. Additionally, a healthy diet might go hand in hand with weight reduction in the case of obesity. Consuming alcohol and smoking have an acute analgesic effect in the short-term, with evidence lacking in the long-term. However, this acute effect is outweighed by other harms on cancer survivors’ general health. Last, informing patients about the benefits of an active lifestyle and reducing a sedentary lifestyle after cancer treatment must be emphasised when considering the proven benefits of physical activity in this population. A multimodal approach addressing all relevant lifestyle factors together seems appropriate for managing comorbid conditions, side-effects, and chronic pain after cancer. Further research is needed to evaluate whether modifiable lifestyle factors have a beneficial influence on chronic pain among cancer survivors.

## 1. Introduction

Cancer has overtaken vascular diseases as the leading cause of death in high-income countries [1]. On top of that, it is expected that the global cancer burden will grow 47% by 2040 [2]. Despite these appalling numbers, cancer survivorship has fortunately increased to 70% in developed countries, mainly due to early detections and treatment advances [3]. 

Different definitions for cancer survivor (CS) exist, but according to a systematic review of Marzorati et al., (2017), the most widely used definition is: “being a CS, starts on the day of diagnosis and continues until the end of life” [4]. Three cancer survivorship phases can be distinguished: “acute survivorship” (i.e., early-stage or time during curative treatment), “permanent survivorship” (i.e., living with cancer or also called the palliative stage), and “extended survivorship” (i.e., cured but not free of suffering) [4]. This article focuses on the extended survivorship phase since it is difficult for cancer survivors (CSs) to recognize themselves as ‘cured’ if they continue to suffer after treatment completion [4]. Unfortunately, in this phase, an important proportion of these CSs will face unwanted and debilitating adverse effects that arise or persist beyond primary treatment, which is frightening and should therefore be dealt with seriously [5]. 

Chronic pain is one of these and occurs in 40% of CSs [6]. Chronic pain is defined by the International Association for the Study of Pain (IASP) as pain that persists or recurs for longer than three months [7]. Unrelieved pain can have considerable adverse consequences on a CSs’ quality of life [6]. Therefore, providing CSs with optimal pain treatments is essential to reduce their psychological, physical, and socio-economic impact [6]. Although several initiatives attempted to increase awareness about (post) cancer pain (e.g., the Global Year Against Cancer Pain in 2008 promoted by IASP), chronic pain in CSs remains undertreated, misunderstood, and highly prevalent [6].

Nowadays, the National Comprehensive Cancer Network guidelines [8] advise pharmacological and non-pharmacological treatments for pain during cancer treatment, but after treatment, a decrease of pain medication is recommended to avoid the risk of addiction, misuse, and adverse effects such as opioid-induced hyperalgesia and sleeping disruptions. Unfortunately, shifting towards non-pharmacological treatments remains challenging for many oncologists since they are used to treat patients with acute pain associated with cancer or its therapy [9]. However, the aggressive and curative treatments, including surgery, chemo-, radio- and or maintenance therapy, are not the only factors contributing to the transition of acute to chronic pain. Other factors such as young age at diagnosis, depression, anxiety, low education, and negative lifestyle behaviour (e.g., high body mass index (BMI), low physical activity levels, high alcohol consumption, etc.) might have an impact as well [10,11,12]. Unfortunately, not all these factors are treatable or modifiable. However, new evidence on healthy lifestyle behaviour demonstrates promising results on pain, quality of life, cancer recurrence, psychological well-being [13,14,15,16]. A healthy lifestyle is defined as actions or method one initiate to achieve optimum health and lower the risk of disease or early death [17], which underlines the need to target (pain) multimodally and tailor treatment according to the CS’s needs [18]. Therefore, the purpose of this paper is to review and update knowledge on chronic pain and modifiable lifestyle factors in CSs and to discuss the beneficial impact of modifiable lifestyle factors on chronic pain after cancer (Figure 1).

## 2. Methods

The best evidence regarding lifestyle behaviour and chronic pain in CSs was retrieved in PubMed and Web of Science up to September 2021. Relevant articles were selected by combining the following keywords: CS, chronic pain, lifestyle factors, risk factors, smoking, dietary intake, physical activity, obesity, medication, distress, stress, sleep disorders. To be included, articles had to meet the following criteria: (1) display original data in CSs; (2) address the aims of this review; (3) be published as full articles; and (4) written in English, Dutch, German or French. The following criteria were applied for exclusion: (1) articles reporting animal studies; and (2) studies with the following study design: case reports, congress proceedings, abstracts, letters to the editor, opinions or editorials.

## 3. State-of-the-Art

### 3.1. Pain

Chronic cancer-related pain represented in the International Classification of Diseases (ICD-11) differs from the pain of other chronic pain populations [19]. Chronic pain in CSs is caused by damage of primary cancer, its metastasis or its treatment, inducing chronic secondary pain syndromes such as musculoskeletal and neuropathic pains [7]. That can persist over time if no adequate pain management was provided initially [7].

Glare et al., (2014) published a comprehensive overview of the types of treatment-related cancer pain arising after the curative treatments [19]. For example, post-operative syndromes might occur after surgery, such as phantom pain after amputation, post-mastectomy pain and other complications [19]. Furthermore, chemo- and radiotherapy can also cause adverse effects. Chemotherapy, for example, can cause symmetrical painful numbness, burning, and tingling in both hands and feet. On top of that, it could also lead to osteoporosis, osteonecrosis, arthralgias, and myalgia. Radiotherapy can lead to serious adverse effects caused by ionising radiation, inducing reactive oxygen species (ROS) production, and DNA and regulatory proteins damage to targeted cells. These provoke apoptosis and increased inflammation in the exposed cells and the neighbouring cells by radiation-induced bystander effects, possibly leading to plexopathies and osteoradionecrosis [19,20]. Maintenance therapy like aromatase inhibitors can produce arthralgia and myalgia [19]. In addition to these adverse effects, health care providers have to evaluate new arising or aggravating pain complaints with caution because these can indicate a recurrence or a second malignant tumour [19]. 

Despite the existing guidelines, chronic pain remains underrecognized and mistreated in the extended survivorship phase [5]. Under recognition might be due to: (1) patients’ belief that pain is inevitable and uncontrollable, causing them not to report pain to their physicians; and/or (2) physicians’ poor knowledge of pain assessment methods [21]. Mistreatment of pain, on the other hand, might be due to: (1) suboptimal communication between CSs and physicians; (2) non-adherence of the patients due to misconception of pain medication; and/or (3) lack of knowledge or confidence of the physicians in applying pain management guidelines in the clinical field [22]. Moreover, CSs typically are insufficiently informed about the origin of their pain, the possibilities of pain relief, and how they can access support when needed, which might affect their happiness of having survived and beaten cancer [23,24,25]. 

Over the last decade, the education provided to CSs made a shift from a biomedical pain management, falling short in explaining persistent pain, to a biopsychosocial pain management [26]. This is in concordance with recent findings of the multidimensional aspect of pain [23]. Psychosocial factors, such as cognitive appraisals and expectations, are cornerstones in the patient’s pain experience and might bring patients in a downward spiral if not considered [27]. The underlying mechanism can be explained by the fact that psychological factors and pain sensations share similar brain activity, such as the prefrontal cortex, thalamus, hypothalamus, and amygdala and might subsequently affect the descending nociceptive pathways of the periaqueductal grey and rostro-ventral medulla [28]. So, depressive mood, anxiety, and cognitions play an essential role in pain modulation, and the understanding of its mechanism is primordial for appropriate assessment and treatment [10,28]. One cognitive appraisal that gained attention in the past years is perceived injustice (PI) [29,30]. It is demonstrated that people experiencing PI, attribute blame to others for their suffering, have the tendency to interpret their losses as severe and irreparable, and experience a sense of unfairness [29] (*e.g., someone who never smoked yet was diagnosed with lung cancer*). A systematic review showed significant associations between PI and worse pain-related outcomes, including more intense pain, more disability, and worse mental health [31]. These along with lower quality of life are seen in breast CSs with higher PI scores, and PI rather than pain catastrophizing mediates the relationship between pain and quality of life [32]. A more intense expression in terms of their suffering and loss is seen due to increased maladaptive pain behaviour. In turn, this increases the likelihood of being prescribed opioids [29,33]. People displaying more maladaptive pain behaviour affect clinicians’ decision to prescribe opioids [34]. Considering the known long-term adverse effects of long-term opioid use [9] and the possibility of developing opiate-induced hyperalgesia [35], PI seems to be a new perspective that should be further investigated in the future. 

Other factors that also play a vital role in chronic pain after cancer are associated with patients’ healthy lifestyle behaviour. Addressing modifiable lifestyle factors is essential to prevent recurrence of cancer, adverse effects, mortality, as well as improving quality of life and pain relief [36,37]. These factors’ impacts and their relationship with pain in CSs are discussed in detail in the following sections of this paper (Figure 2).

### 3.2. Lifestyle Behaviour

#### 3.2.1. Stress

Stress has been categorised as “the health epidemic of the 21st century” by the World Health Organization (WHO) [38]. It has been defined as a state, whether an actual or perceived event disturbs the physiological homeostasis or the psychological well-being [39,40]. About 12.6% of CSs will develop a lifetime cancer-related post-traumatic stress disorder [41]. Additionally, during survivorship, a substantial proportion of CSs are confronting lingering adverse events and/or experiencing an intense fear of recurrence, both causing anxiety and major distress [42]. Cancer-related distress is defined as a state during which CS cannot deal with their cancer, treatment, or adverse effects due to interference of a multifactorial unpleasant psychological, social, spiritual, or physical event. Distress can transfer normal feelings to disabling problems such as panic attacks, depression, anxiety, existential crises [43]. The presence of chronic stress or distress sustains the overproduction of pro-inflammatory cytokines, which in turn induces fatigue, sleep disorders, depression, and symptoms of sickness [44]. The other stress-related mechanisms behind a heightened inflammation level are higher stress-induced sympathetic activity or a dysregulated hypothalamic-pituitary-adrenal axis (and associated cortisol dysbalance as a characteristic feature of long-term stress exposure) [44,45]. New insights also point out that distress in CSs changes the function and/or structure of some areas of the brain, such as the thalamus, amygdala, prefrontal cortex, hippocampus, subgenual area, hypothalamus, basal ganglia and insula, which are mainly the same areas associated with chronic pain [28,46]. Understanding these changes may open new treatment perspectives and enhance the quality of provided interventions for distress among CSs. 

Early screening of distress might enhance treatment response [42,47]. As stated in the systematic review of Syrowatka et al., (2017), several predictors for distress after cancer could be identified according to the provided treatment, sociodemographic characteristics, comorbidities, and modifiable lifestyle factors (Table 1, Figure 2) [42]. Interestingly, pain is one of the manageable risk factors for distress creating a problematic back loop because distress, in turn, promotes pain by dysregulating the autonomic, endocrine, and immune response [44,48]. This vicious cycle can be interrupted by cognitive behavioural stress management (CBSM) consisting of aspects of cognitive behavioural therapy (CBT) [49,50,51] or, more precisely, coping skills for stress management combined with relaxation training [45,52,53,54]. According to recent published systematic reviews and meta-analyses, CBT has a beneficial effect on cortisol secretion, distress, anxiety, depression, emotional well-being, and negative thoughts in CSs [49,50,51]. Mindfulness-based stress reduction (MBSR) and yoga have also shown promising results on distress in CSs (Figure 2) [52,53,54].

#### 3.2.2. Sleep

Insomnia is one the most frequently experienced survivorship concerns and is characterised by difficulty with sleep initiation, duration, consolidation, and quality, resulting in daytime impairments and distress. These difficulties have to occur at least three times a week for more than one month [55]. Insomnia affects more than 30% of CSs years after treatment ending [56,57,58]. The two-fold higher prevalence rate in comparison to the general population can be attributed to the emotional consequences of cancer diagnosis, the direct effects of cancer treatment, and its side-effects [56]. Among cancer patients, prevalence numbers of insomnia are the highest in breast and gynaecologic cancers compared to prostate cancer [56]. Breast CSs are particularly vulnerable to insomnia due to fear of recurrence, endocrine therapy, and other hormonal changes related to breast cancer treatment [59,60,61]. Due to hormonal changes, about 85% of breast CSs will report hot flushes, night sweats and arthralgia, resulting in multiple awakenings throughout the night [62,63]. Moreover, breast CSs with hot flushes and (joint) pain are respectively 2.25 (95% CI 1.64–3.08) and 2.31 (95% CI 1.36–3.92) more likely to develop sleep problems (Table 1, Figure 2) [64]. On the other hand, in non-cancer populations, insomnia forms a higher risk for developing future chronic pain disorders compared to chronic pain leading to new insomnia cases [65]. Sleep problems lower pain thresholds and exacerbate response to painful stimuli by dysregulating the immune system, hypothalamus-pituitary-adrenal axis, monoaminergic pathways, and endogenous substances (adenosine, nitric oxide, melatonin, and orexin), which will, for example, increase the pro-inflammatory state [66].

Based on compelling efficacy data, CBT for insomnia (CBT-I) is the gold standard treatment for insomnia (Figure 2) [67]. CBT-I addresses cognitive and behavioural factors that perpetuate insomnia using a multi-component treatment that includes sleep hygiene, stimulus control, sleep restriction, cognitive therapy and relaxation training [68]. The efficacy of CBT-I in CSs was investigated by a systematic review of Johnson et al., (2016) [57] in which they demonstrated that CBT-I improves insomnia symptom severity, sleep efficiency, sleep onset latency, and wake after sleep onset in CSs. The same research question was investigated specifically in breast CSs by a recent review of Ma et al., (2021) [69], in which moderate to large treatment effects were found with clinically significant effects lasting up to one year after therapy for insomnia symptom severity, sleep efficiency and sleep onset latency. Even though solid evidence has shown that CBT-I improves sleep in CSs [57], it remains underused and not readily available in the community or clinical settings [70]. Barriers on the provider level are a shortage of CBT-I specialists and a lack of physician training about sleep [71,72]. On the patient level, barriers include limited understanding of the consequences of insomnia, limited awareness of available treatment options and lack of treatment adherence due to the possible burdensome treatment format [73,74]. There is no doubt about the effectiveness of CBT-I in CSs. However, future studies are needed to investigate the optimal integration of the CBT-I components before adding to the pain management.

#### 3.2.3. Diet

##### Dietary Intake 

Dietary recommendations have only recently been brought into the picture for CSs treatment; therefore, the literature is sparse and limited to breast CSs. However, nutritional guidelines have been introduced by the National Cancer Institute, American Cancer Society, Academy of Nutrition to encourage CSs to start a healthy and prudent diet [13,75]. Unfortunately, the adherence is low because CSs have no guarantee that their prognosis will improve by adopting a healthy diet [76]. According to a meta-analysis of cohort studies, a Western diet, which is characterised by a high consumption of eggs, red meats, and processed foods, is associated with a higher risk of mortality (odds ratio = 1.51; 95% CI 1.24–1.85) and cancer recurrence (odds ratio = 1.34; 95% CI 0.61–2.92) in CSs [77]. However, weak evidence suggests that CSs may be able to reduce their mortality and cancer recurrence rate by switching to a healthy diet that consists of fruits, vegetables, fish, and whole grains after diagnosis [78]. A healthy diet is usually rich in anti-oxidative, anti-inflammatory, endothelial protective, metabolic substances, which affect tumour growth and promote cancer apoptosis [79]. As advised by different associations, nutritional counselling should be provided by registered dietitians specialised in oncology [13]. 

Furthermore, ongoing research shows that food could have both an adverse and a beneficial influence on chronic pain. A recent systematic review revealed that studies examining whether diet influences chronic pain in CSs are essentially lacking (Table 1) [80]. Nevertheless, evidence in breast CSs points out some significant relation between pain and nutrition. A network meta-analysis for therapeutic options for aromatase inhibitor-associated arthralgia in breast cancer has suggested that omega-3 fatty acids might be effective in reducing pain severity scores and pointed out the need for further evaluation for omega-3 fatty acids as well as vitamin D (Table 1) [81]. Additionally, a cross-sectional study showed clearly that breast CSs who were well-nourished or anabolic according to category A of the patient-generated subjective global assessment (PG-SGA) had fewer pain symptoms than those who were malnourished category B of PG-SGA [82].

As discussed earlier, nutritional sciences are only now beginning to address chronic pain in CSs. However, why should “diet” be advised in chronic pain management to CSs? Knowing the benefits and drawbacks of various diets for survivors with chronic pain could be the key to finding a clear answer. The most important vision of implementing a specific diet in pain management is based on using regulatory effects of nutrition on several pain mechanisms with no or bare minimum side effects. This could provide a long-term, sustainable, and cost-effective pain management alternative for CSs. Therefore, in the future, interdisciplinary collaboration across researchers and clinicians is needed to unravel the role of nutrition in pain-related mechanisms and its implications on pain reduction in CSs. Currently, the lack of evidence supporting the added value of dietary interventions for chronic pain management in CSs precludes to advise its use (Figure 2). 

##### Obesity

Obesity is a condition characterised by an increase in body fat [83,84]. At the neurobiological level, obesity is considered to cause pain through various mechanisms, including inflammation and hormone imbalance [85]. At the mechanical level, obesity can also cause pain by structural overloading [84,86], which can lead to altered body posture and joint misuse [87]. The latest review in taxane- and platinum-treated CSs demonstrated a good-to-moderate relationship between obesity and higher severity or incidence of chemotherapy-induced peripheral neuropathy (CIPN), with moderate evidence showing diabetes did not increase incidence or severity of CIPN [88]. Furthermore, a systematic review with meta-analyses of Leysen et al., (2017) demonstrated that breast CSs with a BMI > 30 have a higher risk (odds ratio = 1.34, 95% CI 1.08–1.67) of developing pain (Table 1, Figure 2) [12]. However, more research is needed to determine the long-term impact of obesity among the expanding population of CSs [89]. Studies looking at the link between changes in body mass index, fat mass, inflammatory markers, and chronic pain might help us better comprehend the relationship between these variables in the CS population. Additionally, well-designed, high-quality randomised controlled trials on the effect of combined weight loss/pain therapies are required to inform patients and clinicians on how to personalise the approach to reduce chronic pain prevalence, intensity, or severity in CSs through obesity management (Figure 2).

#### 3.2.4. Smoking

Smoking tobacco and, to a lesser extent, e-cigarettes is well-known to negatively influence cancer’s prognosis and forms a major risk factor for various cancer types and several other chronic diseases [90,91,92]. Smoking cessation has a favourable effect on treatment efficacy, psychological well-being and general quality of life [93]. The National Comprehensive Cancer Network offers a guideline for smoking cessation, consisting of pharmacotherapy (e.g., nicotine replacement therapy or varenicline) and behaviour therapy (Figure 2) [47,94]. This program is more successful when initiated at the time of diagnosis because an early start avoids more adverse effects [90]. Patients who continue to smoke have a higher likelihood of facing post-operative complications due to (wound) infections, failed reconstruction and tissue necrosis, which could lead to prolonged hospitalisation [95,96]. Unfortunately, a big proportion of young CSs continue to smoke after their diagnosis. Approximately 25.2% of CSs aged 18 to 44 years were current smokers compared to 15.8% in the general population [97]. Thus, during the survivor phase, additional support should be provided to target patients’ barriers to smoking cessation to prevent cancer recurrence. 

Pain might be one of the barriers to smoking cessation in CSs [98]. An observational study by Aigner et al., (2016) demonstrated that when patients experience higher pain levels, they usually smoke a larger number of cigarettes during these days and initiate fewer attempts to quit smoking [98]. This can be explained by the fact that nicotine produces an acute analgesic effect, making it much harder for them to stop due to the rewarding sensation they experience [99]. Despite its short-term analgesic effect, tobacco smoking sustains pain in the long-term [93]. This underlines the importance of incorporating anti-smoking medications in CSs with pain to avoid relapse during nicotine withdrawal [99]. Moreover, pain management should be added to the counselling aspect to enhance the patient’s knowledge, which in turn, might improve their adherence to the whole smoking cessation program [98]. Furthermore, the 5As (Ask, Advise, Assess, Assist, Arrange) approach, which assesses the willingness of the patient to quit smoking, is no longer recommended since studies have demonstrated that smokers who did not feel ready to quit smoking at the same rate as those who wanted to [100]. The model with the most promising results might be “opt-out”, during which health care providers offer counselling and pharmacotherapy to all smokers, which is more ethical [101]. However, research on how to integrate this approach in current cancer care for CSs is needed. 

#### 3.2.5. Alcohol Consumption

Similar to smoking, alcohol consumption is a preventable risk factor for liver, oesophageal, colorectal, breast, head, neck, and many other cancers [102]. It is established that excessive or binge drinking enhances the likelihood of cancer recurrence, bad prognosis, or death [77]. Despite this, up to now, no evidence supports or refutes that drinking with moderation (≤1 drink for women and ≤2 drinks for men per day) is associated with a lower risk of cancer [103,104,105]. On top of this, some studies show a reduction in risk due to moderate alcohol intake, which might be explained by confounders, and/or the anti-cancer effect of polyphenols (present in wine) [106] or phytoestrogen and polysaccharides (present in beer) that lower free testosterone, inducing prostate cancer [107,108]. However, these small benefits are quickly outweighed by other harms of alcohol consumption. Furthermore, a growing trend in alcohol intake among CSs is observed, but no explanation for this trend could be found [109]. Nevertheless, alcohol consumption can initiate people to smoke or smoke even more [109]. Combining both multiplies their adverse effects because alcohol slows down the body’s capacity to eliminate the carcinogenic chemicals of smoking [97,109,110]. These findings highlight the importance of increasing CSs’ awareness about these lifestyle factors. 

The impact of alcohol use on pain is poorly investigated in CSs, but according to one systematic review of two cohort studies, the risk of developing pain can be reduced by alcohol use (Table 1) [12]. This finding might be misleading due to the fact that alcohol has an acute analgesic effect [111]. In non-cancer populations, studies demonstrated that this analgesic effect diminishes over time, and there is an association between chronic pain and alcohol consumption [112]. This pain might be evoked by developing alcoholic neuropathy, musculoskeletal disorders, or alcohol withdrawal [112]. Conversely, chronic pain increases the risk of alcohol abuse [113]. Nevertheless, psychosocial factors are also highly present in patients with alcohol abuse and can be attributed to abnormalities in the reward system of the brain [114]. Additionally, a recently published study demonstrated that chronic pain patients with high levels of pain catastrophising are more likely to be heavy drinkers [115]. General advice on alcohol consumption after cancer is currently not possible due to the high variability of results in different CSs. Therefore, health care providers should tailor their advice according to cancer types and patients [116]. Within that view, an overview of recommendations regarding individualised alcohol consumption for each CS type could support clinicians in doing so, yet such evidence-based recommendations are currently lacking (Figure 2).

#### 3.2.6. Physical Activity

Being physically active after a cancer diagnosis improves CSs’ survival rate by 30% [117,118,119], which underlines that healthy behaviour during the extended survival phase is essential [117]. The American College of Sports Medicine, American Cancer Society and the US Department of Health and Human Services developed exercise guidelines that advise every CS to engage weekly in 75 min of vigorous-intensity or 150 min of moderate-intensity aerobic physical activity [90,120,121]. For instance, the evidence demonstrated that supervised physical activity reduces cancer-related fatigue, depression, and increases quality of life, cardiovascular and musculoskeletal fitness in CSs [14,15,16]. Additional beneficial effects of physical activity were also seen on musculoskeletal pain and stiffness in breast CSs taking aromatase inhibitors for a long period (Table 1, Figure 2) [81,122,123]. However, only few CSs attain the recommended physical activity levels, with pain being an important limiting factor [116,124]. Inappropriate beliefs regarding the expected outcome of physical activity represent a major barrier for CSs to engage in physical activity programs. For example, some breast CSs fear that resistance exercises can aggravate cancer-related lymphedema, which is proven to be wrong as resistance exercises are perfectly safe in this group and do not increase lymphedema [125], others might fear that exercise can exacerbate their pain, which was refuted by systematic reviews with meta-analyses in CSs and a Cochrane review in chronic non-cancer pain populations, demonstrating that physical activity has a small positive effect on pain (Table 1, Figure 2) [123,126,127]. Despite all this evidence, patients’ adherence to physical activity remains low and remains a bottleneck in current care [128]. Therefore, how to reduce a sedentary lifestyle in CSs with chronic pain should be more thoroughly investigated and implemented in guidelines, and patients should be better informed about the benefits of an active lifestyle [128].

Identifying predictors of adherence will offer the possibility to provide personalised guidance to CSs who are less likely to adhere to exercise, which will undoubtedly lead to better treatment outcomes [129]. According to a systematic review, behavioural (i.e., motivation) and sociodemographic predictors (i.e., distance and social support of the family or therapists) should be addressed [130]. To improve CSs’ exercise motivation or lifestyle behaviours, motivational interviewing can be used [131]. During this patient-centred approach, five different stages can be distinguished: pre-contemplation, contemplation, preparation, action, and maintenance. In each stage, behaviour changes will be tackled differently [130,131]. A Cochrane review concluded that exercise interventions with determined goals, graded activity, and behaviour change reached the highest adherence in CSs [118]. Behavioural graded activity is such an intervention that combines these three components and aims (i.e., determined goals, graded activity, and behaviour change) to target patients’ difficulties and complaints during their daily living [132]. This approach might enhance patients’ willingness to adhere to healthy behaviour compared to other exercise interventions. Additionally, in recent years, alternative therapies such as mindfulness-based approaches, hypnosis and yoga gained importance and demonstrated significant beneficial effects on quality of life, psychological distress, anxiety, depression, fear of cancer recurrence, fatigue, sleep, and pain [133,134,135]. Obviously, mindfulness-based approaches and yoga fit into the ‘stress management’ category as well, and therefore potentially serve two lifestyle factors (i.e., stress and physical therapy). However, more research is needed to find the optimal approach for higher long-term adherence to an active lifestyle in CSs.

**Table 1 jcm-11-00195-t001:** Evidence of lifestyle factors on pain in cancer survivors. Abbreviations: AIA: Aromatase Inhibitor-associated Arthralgia; C: Cohort; CI: Confidence Interval; CIPN: Chemotherapy-Induced Peripheral Neurotoxicity; CS: Cross-sectional Study; ES: Effect Size; I^2^: Heterogeneity; MD: Mean Difference; OR: Odds Ratio; *p*: *p*-value; RCT: Randomized Controlled Trial; SMD: Standardized Mean Difference; SORT: Strength of Recommendation Taxonomy.

Lifestyle Factor	First Author, Year Published, Study Type	Included Population	Number of Included Studies (n_1_) and Participants (n_2_)	Detail of Lifestyle Factor/Intervention Assessed	Main Results in Context of the Specified State-of-the-Art	Level of Evidence [136]
Alcohol consumption	Leysen et al., 2017, Systematic review with meta-analysis [12]	Breast Cancer Survivors	n_1_ = 2 (1 CS and 1 C) and n_2_ = 2519	Alcohol use	Alcohol (OR 0.94, 95% CI [0.47, 1.89], *p* = 0.86, I^2^ = 67%) was not a predictor for pain, Inconsistent and low evidence	3b
Diet	Kim et al., 2018, Systematic review of systematic reviews [81]	Breast Cancer Survivors with AIA	n_1_ = 3 (systematic review of RCT), and n_2_Omega-3_ = 817, and n_2_VD_ = 453	Omega-3 Fatty Acids, and Vitamin D	Significant effects were found for omega-3 fatty acids (MD −2.10, 95% CI [−3.23, −0.97]), and vitamin D (MD 0.63, 95% CI [0.13, 1.13]) on pain, Low evidence	1a
	Yilmaz et al., 2021, Systematic review [80]	Cancer Survivors	n_1_ = 2 (uncontrolled clinical trial) and n_2_ = 77	Nutritional supplements: vitamin C, chondroitin, and glucosamine	Lack of evidence	2a
Obesity	Leysen et al., 2017, Systematic review with meta-analysis [12]	Breast Cancer Survivors	n_1_ = 7 (4 CS and 3 C) and n_2_ = 5573	BMI	BMI > 30 (OR 1.34, 95% CI [1.08, 1.67], *p* = 0.008, I^2^ = 33%,) was a predictor for pain, Consistent and low evidence	3b
	Timmins et al., 2021, Systematic review [88]	Cancer Survivors	n_1_ = 16 (3 CS, 11 C, and 2 retrospective chart review) and n_2_ = 14,033	Obesity	According to the SORT: the association between obesity and CIPN was good-to-moderate patient-centred evidence	3b
Physical Activity	Boing et al., 2020, Systematic review with meta-analysis [123]	Breast Cancer Survivors with AIA	n_1_ = 3 (2 RCT, 1 pilot study), and n_2_ = 118	Exercise	Significant effect was found on pain (SMD −0.55, 95 % CI [−1.11, −0.00], *p* = 0.05 I^2^ = 80%), Low Evidence	1b
	Kim et al., 2018, Systematic review of systematic reviews [81]	Breast Cancer Survivors with AIA	n_1_ = 2 (systematic review of RCT), and n_2_ = 262	Aerobic Exercise	No significant effect was found on pain (MD −0.80, 95% CI [−1.33, 0.016]), Low evidence	1a
	Lavín-Pérez et al., 2021, Systematic review with meta-analysis [127]	Cancer Survivors	n_1_ = 7 (RCT), and n_2_ = 355	Exercise (HIT)	Significant effect was found on pain (SMD −0.18, 95% CI [−0.34, −0.02], *p* = 0.02, I^2^ = 4%), Moderate evidence	1a
	Lu et al., 2020, Systematic review with meta-analysis [122]	Breast Cancer Survivors with AIA	n_1_ = 6 (RCT), and n_2_ = 416	Exercise	Significant effect was found on pain (SMD −0.46, 95% CI [−0.79, −0.13], *p* = 0.006, I^2^ = 63%), Moderate evidence	1a
	Timmins et al., 2021, Systematic review [88]	Cancer Survivors	n_1_ = 5 (2 C and 3 CS), and n_2_ = 3950	Low physical activity	According to the SORT: the association between physical inactivity and CIPN was of moderate evidence	3b
Sleep	Leysen et al., 2019, Systematic review with meta-analysis [64]	Breast Cancer Survivors	n_1_ = 4 (2 CS and 2 C) and n_2_ = 1907	Sleep Disturbances	Pain was a predictor for sleep disturbances (OR 1.68, 95% CI [1.19, 2.37], *p* = 0.05, I^2^ = 55%, after subgroup analysis OR 2.31, 95% CI [1.36, 3.92], *p* = 0.002, I^2^ = 27%)	3b
Smoking	Leysen et al., 2017, Systematic review with meta-analysis [12]	Breast Cancer Survivors	n_1_ = 2 (1 CS and 1 C) and n_2_ = 2519	Smoking status	Smoking (OR 0.75, 95% CI [0.62, 0.92], *p* = 0.005, I^2^ = 0%) was not a predictor for pain, Consistent and low evidence	3b
Stress	Syrowatka et al., 2017, Systematic review [42]	Breast Cancer Survivors	n_1_ = 12 (6 CS and 6 C) and n_2_ = 7842	Distress	Pain was significantly associated with distress: 9/12 studies (75%)	3b
Intervention	Chang et al., 2020, Systematic review with meta-analysis [54]	Breast Cancer Survivors	n_1_ = 5 (RCT) and n_2_ = 827	Mindfulness-Based interventions	No significant effect was found on pain (SMD −0.39, 95% CI, [−0.81, 0.03], *p* = 0.07, I^2^ = 85%), Moderate evidence	1a
	Cillessen et al., 2019, Systematic review with meta-analysis [133]	Cancer Patients and Survivors	n_1_ = 4 (RCT) and n_2_ = 587	Mindfulness-Based interventions	Significant effect was found on pain (ES 0.2, 95% CI [0.04, 0.36], *p* = 0.16, I^2^ = 0%), Moderate evidence	1a
	Martinez-Miranda [26]	Breast Cancer Survivors	n_1_ = 2 (RCT) and n_2_ = 134	Patient Education	No significant effect was found on pain (SMD −0.05, 95% CI [−0.26, 0.17], *p* = 0.67, I^2^ = 0%, Low evidence	1a
	Silva et al., 2019, Systematic review [137]	Cancer Survivors	n_1_ = 4 (4 quasi-experimental studies), and n_2_ = 522	Promoting healthy behaviour by mHealth apps	Effect found on pain was inconsistent and of low quality of evidence	2b

## 4. Future Directions for Scientists

First, it is recommended that researchers make a clear distinction between CSs’ phases when initiating and reporting studies in CSs. Currently, the term CS is too globally used, making it difficult to compare or combine results of studies due to their high heterogeneity. An individual in palliative care has different needs than an individual that is cured of cancer; however, both are CSs according to the most widely used definition [4]. A distinction between the different phases has been described by Mullan et al., in 1985 [138]. Unfortunately, these terms are not frequently used in the literature [138] even though a clear distinction between phases could help clinicians to communicate more easily and to provide the appropriate care to patients’ needs according to their phase in the survival of cancer. 

Second, most studies were performed on Caucasian breast CSs with high socio-economic status. This population is more likely to have a higher adherence and willingness to change their lifestyle habits [139]. However, to reach a better understanding of barriers for lifestyle changes, research needs to be performed among CS populations with diverse socio-economic backgrounds. This way, oncological care for CSs can be more tailored to patients of different gender, race, and socio-economic capacities.

Third, future studies regarding lifestyle factors in CSs should more thoroughly account for possible confounders. Indeed, research studying a particular lifestyle factor should not only be adjusted for age, gender, education, and so forth, but also for other established lifestyle factors, which might be a considerable confounder. Furthermore, the effects of lifestyle factors in CSs are most often observed over a short period, preventing to draw conclusions regarding long-term impact of lifestyle factors in CSs. More research is warranted to observe the long-term effects of pain management and healthy lifestyle interventions in CSs.

## 5. Future Directions for Clinicians

The literature indicates that implementing healthy lifestyle habits in CSs has low compliance rates [140]. A barrier that might cause low adherence to healthy lifestyle behaviours is the burdensome treatment format of most behavioural interventions [73,74]. Therefore, stepped care models might provide clinicians with a possible solution to improve the feasibility and deliver care efficiently [141]. In existing stepped care models, the first step is typically a form of self-management therapy (e.g., recommendations) with the possibility to progress to the highest step of six to eight individual sessions with a specialist, if needed [142,143]. For example, a recent study in CSs demonstrated that more than 50% of CSs with insomnia benefit form a one-hour group-delivered session that empowers CSs by teaching them about sleep health and provides specific information on how to adapt their sleep behaviours [142]. Interestingly, they found that CSs who had experienced sleep problems for a shorter period and perceived less burden from their sleep problems were most likely to benefit from the one-hour program, suggesting that it is crucial to identify CS with sleep problems as soon as possible to enhance the efficacy of low-intensity interventions [142]. However, further research is warranted before implementing stepped care for the other lifestyle factors. In addition, systematic reviews demonstrated promising findings for virtual therapy, suggesting that virtual interventions might be a possible option to enhance access to care, which solves the distance issue [69,137,144]. 

Furthermore, to reduce the treatment burden, clinicians should perform early screenings and identify negative predictors to improve patients’ self-efficacy to sustain a healthy lifestyle. Developing evidence-based guidelines, including algorithms with practical triage and referral plans to other healthcare professionals, will improve survivorship care. Enhancing the productivity of oncological care by 2025 is of utmost importance because there will be a shortage of oncologists due to the growing cancer population [145]. Besides that, many clinicians have difficulties providing the ideal pain management plan and delivering health promotion guidance due to a lack of knowledge [22]. Supplementary support and educational interventions should be organized for health care providers to enhance their expertise and confidence in this field. 

Another recommendation for future clinical practice is considering the use of pain neuroscience education as a way to decrease the threatening nature of pain, catastrophic thinking and fear-avoidance beliefs in CSs [146]. Cancer patients indicate themselves that they have insufficient knowledge regarding pain during or after cancer, what the possibilities of pain relief are and how they can access support when needed [24,25]. When comparing pain knowledge between CSs, healthy controls and caregivers, CSs had the lowest pain knowledge of the three groups [147]. Education about pain is underused in the field of oncology and non-existent in the survivorship phase [148]. Pain neuroscience education can clear the path for more active approaches to pain management, including providing lifestyle interventions. Manuals with guidelines for clinicians on how to explain pain following cancer [146], including accounting for perceived injustice during pain neuroscience education [149], are available to support clinicians in doing so.

Lastly, this state-of-the-art paper underlines once more the complexity of managing chronic pain in CSs. As discussed previously, adopting a healthy lifestyle might have a beneficial influence on the chronic pain of CSs. Unfortunately, there is currently a lack of research about the effectiveness of modifiable lifestyle factors on pain. Moreover, pain in CSs should be targeted on cognitive, behavioural, sensory and emotional levels due to its complexity [18]. Therefore, all pain interventions should be multidisciplinary and personalized for each CS [19].

## 6. Conclusions

Emerging evidence shows that CSs find it challenging to receive optimal treatment plans for their burdens, and support or reinforcement to maintain a healthy lifestyle. Therefore, it is crucially important to prepare clinicians well, so they can provide guidance along and after primary treatment. For chronic pain in CSs, it is primordial to identify factors that contribute to the transition of acute to chronic pain in CSs because chronic pain remains underrecognized and mistreated in this population. Furthermore, a proper definition between CSs’ phases should be developed for optimal research and treatment. In the clinical field, new psychosocial factors and modifiable lifestyle factors should be targeted to improve pain relief in CSs.

Modifiable lifestyle factors and their impact on pain have been discussed in depth in this paper and are, for instance, stress, insomnia, diet, obesity, smoking, alcohol consumption and physical activity. First, an inappropriate stress response promotes pain by dysregulating the autonomic, endocrine, and immune response creating a problematic back loop because pain is a manageable risk factor for distress. The stress response can be managed by CBSM, CBT, MBSR and yoga. Second, sleep and pain also form a vicious cycle (sleep problems exacerbate response to nociceptive stimuli and pain can disturb sleep quality) that CBT-I can break. Third, guidelines recommend prudent diets in CSs. However, more research is needed to unravel the role of nutrition and obesity in CSs. Fourth, alcohol consumption and smoking are both negative lifestyle behaviours that impact patients’ general health. Smoking cessation should consist of behaviour therapy and medication. Last, physical activity demonstrates its beneficial impact in several systematic reviews. However, the adherence is low and new treatment strategies such as motivational interviewing or BGA should be investigated in CSs to increase treatment outcomes in the long-term.

In the future, there will be an insufficient number of professionals (oncologists) due to the growing cancer population [150,151]. Therefore, it is a priority that researchers refine current treatment plans and define the benefits of modifiable lifestyle factors and their impact on chronic pain in CSs.

## Figures and Tables

**Figure 1 jcm-11-00195-f001:**
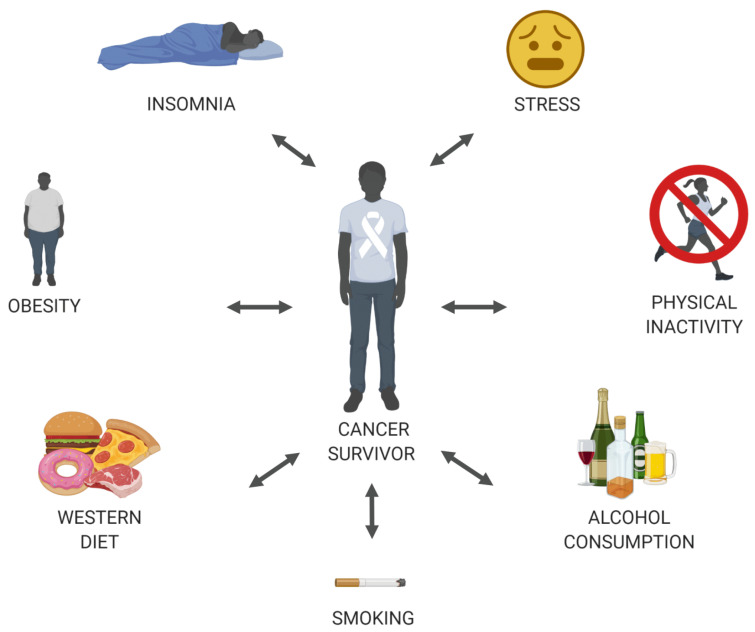
Discussed modifiable lifestyle factors in cancer survivors and might contribute to chronic pain after cancer (Creates with BioRender.com (accessed on: 26 November 2021)).

**Figure 2 jcm-11-00195-f002:**
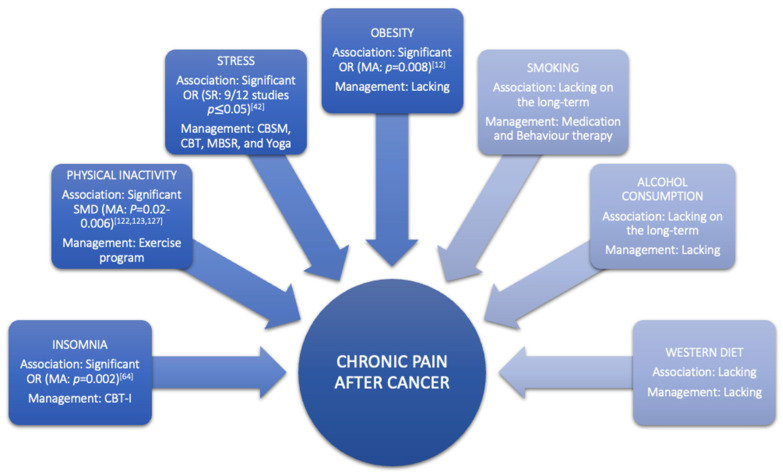
Evidence of modifiable lifestyle factors contributing to chronic pain in cancer survivors. Abbreviations: CBT(-I): Cognitive behavioural therapy (Insomnia); CBSM: Cognitive Behavioural Stress Management; MA: Meta-analysis; MBSR: Mindfulness-based Stress Reduction; OR: Odds Ratio.

## Data Availability

Not applicable.

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
