# Peer review of "Lifestyle and Pain following Cancer: State-of-the-Art and Future Directions"

_jcm, 2021, doi:10.3390/jcm11010195_

Round 1
Reviewer 1 Report
This is an interesting article about Lifestyle and Pain Following Cancer: State of the Art and Future Directions.
This paper is easily readable.
Major suggested revisions (2) are:
- A lack of definitions of some terms or concepts.
Imprecision as to their uses.
Which can give the impression of a popular article.
For example:
L 77 "healthy lifestyle"
L103 "aggressive disease"
L104 "tissue damage"
L163 "Stress"
L182 "Distress"
L196 “Insomnia”
L258 "well-nourished"
L335 "damaging toxins"
- The article is not limited to the problem of pain as suggested in the title. It could be alleviated by removing many parts that are too general or unrelated to pain in survivors.
For example :
unnecessary generalities :
L106 "Researchers have tried to develop new cancer treatment techniques and / or refine old
treatments to reduce undesirable consequences. However, several current cancer treatments still yield adverse effects ”
L165 “However, most of us cannot imagine the physical and psychological stress that goes hand in hand with the life-threatening cancer diagnosis”
Discussions unrelated to Pain Following Cancer:
Figure 2 is a good summary, the article should focus on.
L217
“The efficacy of CBT-I in CSs was investigated by a systematic review of Johnson et al. (2016) [48] in which they demonstrated… with clinically significant effects lasting up to one year after therapy for insomnia symptom severity, sleep efficiency and sleep onset latency
L231
“There is no doubt about the effectiveness of CBT-I in CSs. However, future studies are needed to investigate the optimal integration of the CBT-I components before becoming the first-line treatment for insomnia. ”
L235-L249
“Dietary recommendations have only recently been brought into the picture for CSs 235
treatment; ….. characterised by high consumption of eggs, red meats, and processed foods, is associated with a higher risk of mortality …. A healthy diet is usually rich in anti-oxidative, anti- inflammatory, endothelial protective, metabolic substances, which affect tumour growth
and promote cancer apoptosis. ….”
L289-L301
“Smoking tobacco and, to a lesser extent, e-cigarettes is well-known to negatively influence cancer’s prognosis and forms a major risk factor for various cancer types and …. Thus, during the survivor phase, additional support should be provided to target patients’ barriers to smoking cessation to prevent cancer recurrence.”
L322-L326
“Similar to smoking, alcohol consumption is a preventable risk factor for liver ….These findings highlight the importance to increase CSs’ awareness about these life style factors.”
L356-L363
“…physically active after a cancer diagnosis improves CSs’ survival rate by 30% 356
[108-110], … For instance, the evidence demonstrated that supervised physical activity reduces cancer-related fatigue, depression, and increases quality of life, cardiovascular and musculoskeletal fitness in CSs.”
L431-L448
“The literature indicates that implementing healthy lifestyle habits in CSs has low compliance rates [133]. A barrier that might cause low adherence to healthy lifestyle behaviours is the burdensome treatment format of most behavioural interventions …systematic reviews demonstrated promising findings for virtual therapy, suggesting that virtual interventions might be a possible option to enhance access to care, which solves the distance issue”
Minor suggested revisions :
- L111: give some elements of the physiology of post-radiotherapy pain
- Little referenced or insufficient statements
o L175: stress-induced sympathetic activity or a dysregulated hypothalamic-pituitary-adrenalaxis (and associated cortisol dysbalance as a characteristic feature of long-term stress exposure) ”
o L188 “This loop can be interrupted by cognitive behavioral stress management (CBSM) consisting of aspects of cognitive behavioral therapy (CBT), or more precisely coping skills for stress management combined with relaxation training”
o L200 “. Among cancer patients, prevalence numbers of insomnia are the highest in breast cancer compared to other cancer sites (e.g., gynecologic, prostate, head and neck, urinary, gastrointestinal, etc.)”
o Despite the little data on diet and pain, a chapter is devoted to this theme with a lot of general information
o L276 “The latest review in taxane- and platinum-treated CSs demonstrated a good-to-moderate relationship between obesity and higher severity or incidence of chemotherapy-induced peripheral neuropathy [79].” What about the confounding factors, for example the link between obesity and DNID DNID and chronic pain?
- The criticism of the definition of CS is relevant.
It should happen earlier in the text
Reviewer 2 Report
Thank you for permitting me to review this manuscript
Please provide exact definition of chronic pain and cancer pain
Figure 1 sleep per se is not aligned with other factors , it would be better to change with sleep disturbances.
by the way I do not see much difference between figure 1 and figure 2
line 77 I would suppress "state of the art" and add instead "review and update knowledge on chronic pain
Line 325 this sentence is confusing me there is 2 negative word , would it be "lower risk of cancer ? line 337
conclusion line 479-480
there is not enough data to support this in the conclusion this sentence might be deleted or transferred in the discussion section
Line 505
why there would be a lack of oncologists ? please provide reference , I guess this is because of the increase in the incidence of cancer , if that speculation is true , in some countries I do not think this would be a word wild problem
Is there any reference with regard to hypnosis and CS?
